# The 40 kDa Linear Polyethylenimine Inhibits Porcine Reproductive and Respiratory Syndrome Virus Infection by Blocking Its Attachment to Permissive Cells

**DOI:** 10.3390/v11090876

**Published:** 2019-09-19

**Authors:** Jie Wang, Jie Li, Nana Wang, Qi Ji, Mingshuo Li, Yuchen Nan, En-Min Zhou, Yanjin Zhang, Chunyan Wu

**Affiliations:** 1Department of Preventive Veterinary Medicine, College of Veterinary Medicine, Northwest A&F University, Yangling 712100, China; 18829350677@163.com (J.W.); lijie2016c@163.com (J.L.); m17629097868@163.com (N.W.); jiqi10102@163.com (Q.J.); LiMingS1996@163.com (M.L.); nanyuchen2015@nwsuaf.edu.cn (Y.N.); zhouem@nwsuaf.edu.cn (E.-M.Z.); 2Scientific Observing and Experimental Station of Veterinary Pharmacology and Veterinary Biotechnology, Ministry of Agriculture, Yangling 712100, China; 3Molecular Virology Laboratory, VA-MD College of Veterinary Medicine, University of Maryland, College Park, MD 20742, USA; zhangyj@umd.edu

**Keywords:** PRRSV, polyethylenimine, PEI, virion internalization, antiviral, endocytosis

## Abstract

Porcine reproductive and respiratory syndrome (PRRS) is one of the most economically devastating infectious diseases in pigs worldwide. The causative agent is the PRRS virus (PRRSV). In this study, we explored polyethylenimine (PEI), a cationic polymer with different forms (linear or branched), to inhibit the replication of PRRSV. Our results demonstrate that the linear but not the 40 kDa branched PEI, or the 25 kDa linear PEI, were well tolerated in cultured cells and exhibited a broad-spectrum inhibition of heterogeneous *PRRSV-2* isolates in both MARC-145 cells and primary porcine pulmonary alveolar macrophages (PAMs). Further analysis suggests that PEI could prevent the attachment of PRRSV virions to the susceptible cells. Notably, PEI had a minimal effect on PRRSV internalization in MARC-145 cells, whereas PEI promoted the internalization of PRRSV virions in PAMs, which suggests that these two types of cells might have different internalization processes of PRRSV virions. In conclusion, our data demonstrate that PEI could be used as a novel inhibitor against PRRSV.

## 1. Introduction

Porcine reproductive and respiratory syndrome virus (PRRSV) is an enveloped, single-stranded, positive-sense RNA virus, which belongs to the genus *Porartevirus*, of the family *Arteriviridae* [1,2]. The genome of PRRSV is approximately 15 kb in length and contains over 10 open reading frames (ORFs) [3]. The ORF1a and ORF1b of PRRSV account for two-thirds of its genome and encode all non-structural proteins, which are required for viral replication, while ORFs 2–7 encode structural proteins [3]. Based on sequence similarity, all PRRSV isolates within the genus *Porartevirus* were further classified into two species: *PRRSV-1* and *PRRSV-2* [1,2]. *PRRSV-1* and *PRRSV-2* strains share approximately 60% nucleotide sequence identity and exhibit serotype differences [4,5]. PRRSV infection in pigs is highly restricted to cells of monocyte-macrophage lineages, such as pulmonary alveolar macrophages (PAMs) [6,7]. It is believed that PRRSV infection in susceptible cells is mediated by several cellular receptors or factors [8], such as heparin sulfate (HS) [9], vimentin [10], Cluster of differentiation (CD)151 [11], porcine CD163 (CD163) [12], sialoadhesin (CD169) [13], Dendritic Cell-Specific Intercellular adhesion molecule-3-Grabbing Non-integrin(DC-SIGN, also called CD209) [14] and myosin-9 heavy chain(MYH9) [15]. However, only CD163 and MYH9 are indispensable for PRRSV infection [15,16].

Current strategies for PRRS control are inadequate, although substantial efforts have been dedicated to controlling this disease. Since the first report of PRRS in the United States in 1987, PRRS remains one of the major challenges to the global swine industry, while vaccines against PRRS have been commercially available for over two decades [17,18,19]. This suggests the urgent need for novel methods for PRRSV control and prevention, as well as a deep understanding of viral–host interactions between PRRSV and its permissive cells.

Cationic polymers, which have been used as DNA transfection reagents via facilitating the entry of DNA to target cells [20,21], are also known to have potent antimicrobial activity [22,23]. Polyethylenimine (PEI) is one such cationic polymer, and its novel anti-viral activity has been investigated in several studies [24,25,26,27]. Since PEI can be either a linear or branched molecule with a broad molecular weight range, different PEI molecules have great divergence on modulating viral replication. *N*,*N*-dodecyl, methyl-PEI (DMPEI), synthesized from a commercial 750 kDa branched PEI, has been demonstrated to significantly reduce the infectivity of wild-type and drug-resistant influenza A virus [24], as well as HSV-1 and HSV-2 [25]. Meanwhile, the 25 kDa linear PEI is highly effective against human papillomaviruses (HPVs) and human cytomegaloviruses (HCMVs) [26]. However, another report suggests that the 70 kDa branched PEI accelerates HIV-1 infection despite its inhibition of HIV-1 attachment to cells [27]. This paradoxical effect of PEI on HIV infection is not fully understood. The effect of PEI on PRRSV proliferation is not known.

The objective of this study was to explore PEI to inhibit the replication of PRRSV. Our results demonstrate that the 40 kDa linear PEI, but not branched PEI, exhibited a broad-spectrum inhibition of heterogeneous *PRRSV-2* isolates in both MARC-145 cells and primary porcine pulmonary alveolar macrophages (PAMs). Furthermore, we investigated mechanisms of the 40 kDa linear PEI in the inhibition of PRRSV. Our results showed that PEI could reduce the attachment of PRRSV to the susceptible cells. PEI had a minimal effect in the internalization of PRRSV in MARC-145 cells but promoted the internalization of PRRSV into PAMs. This suggests that the internalization process of PRRSV virions into these two types of cells may be different.

## 2. Materials and Methods

### 2.1. Cells, Viruses, and Chemicals

PAMs were collected from 6-week-old PRRSV-negative pigs, as previously described [28]. PAMs were maintained in RPMI 1640 medium (Biological Industries, Beit HaEmek, Israel) supplemented with 10% FBS (Biological Industries). PRRSV-permissive MARC-145 cells were maintained in Dulbecco’s Modified Eagle Medium (DMEM) (Biological Industries) supplemented with 10% FBS as well.

The *PRRSV-2* virus isolates used in this study were VR-2332 (GenBank: EF536003.1), and highly pathogenic PRRSV (HP-PRRSV) isolates SD16 (GenBank: JX087437.1), JXA1 (GenBank: EF112445.1), GD-HD (GenBank: KP793736.1), and NADC30-like isolate HNhx (GenBank: KX766379). All these PRRSV isolates were used to inoculate MARC-145 cells or PAMs at a multiplicity of infection (MOI) as indicated. The median tissue culture infectious dose (TCID_50_) of all the PRRSV isolates (except the NADC30-like isolate HNhx) were titrated in MARC-145 cells as previously described [29]. Propagation and titration of the NADC30-like isolate HNhx were conducted in PAMs.

Three different forms of PEIs were used in this study: a branched PEI and two linear PEIs. The branched PEI (average molecular weight of 75 kDa) was purchased from Sigma-Aldrich (St. Louis, MO, USA) and dissolved in molecular-grade water (Thermo Fisher Scientific, Waltham, MA, USA) at a final concentration of 1 mg/mL. Two linear PEIs (molecular weights of 25 and 40 kDa) were purchased from PolySciences (Warrington, UK) and solubilized in molecular-grade water as well. To make the solution of the 25 kDa linear PEI, the PEI was added to molecular-grade water and then heated to 95 °C until the chemical was completely dissolved. All PEI solutions were sterilized by filtration through a 0.2 μm filter (Millipore, Burlington, MA, USA) and stored at −20 °C before use except the 25 kDa linear PEI solution (stored at room temperature before use to prevent the precipitation of the PEI).

### 2.2. Cell Viability and Cytotoxicity

Cell viability and cytotoxicity were evaluated using the CellTiter-Glo^®^ Luminescent Cell Viability Assay Kit (Promega, Madison, WI, USA) by following the manufacturer’s instructions. Briefly, MARC-145 cells or PAMs were seeded into 24-well plates at a density of 5 × 10^4^ cells/well or 4 × 10^5^ cells/well and cultured for 24 h. Then, fresh medium containing the PEI at the indicated concentration was added to the cells. After incubation for another 24 h, the cells were lysed by using 1× passive cell lysis buffer (Promega). The cell lysate was transferred into a 96-well black polystyrene microplate (Corning Inc, Corning, NY, USA). CellTiter-Glo^®^ reagent was added and mixed with the cell lysate. After incubation for 10 min at room temperature (RT), luminescence signal was determined with VICTORX™ X5 Multilabel Reader (Perkin-Elmer Life and Analytical Sciences, Wellesley, MA, USA).

### 2.3. PRRSV Inhibition Assay

MARC-145 cells were seeded into 24-well plates at a density of 5 × 10^4^ cells/well and cultured for 24 h. The cells were inoculated with a mixture of PRRSV (0.1 or 1 MOI) in plain DMEM and linear PEI or branched PEI (with indicated concentration). The cells were then incubated at 37 °C for 5 h. The mixture containing PEIs and virions was discarded and the cells were rinsed with phosphate-buffered saline (PBS), pH7.2, followed by culturing for another 24 h before further experiments. Subsequently, cells were rinsed with PBS twice and prepared for immunofluorescence assay (IFA), or harvested for Western blot and qPCR analyses. Cell culture supernatant containing progeny virus were further titrated to evaluate the infectious virions as well.

To evaluate the antiviral activity of PEIs in PAMs, cells were seeded into 24-well plates at a density of 5 × 10^5^ cells/well and incubated for 6 h before further experiments. A fresh RPMI 1640 medium containing PRRSV (0.01 MOI) and PEI was added to the cells. After 1 h inoculation, PAMs were rinsed with PBS to remove the virus and PEI mixture unbosund and cultured for another 24 h before further analysis with qPCR, Western blot, and virus titration.

### 2.4. Immunofluorescence Assay (IFA)

MARC-145 cells in cell culture plates were fixed with 4% paraformaldehyde (Sigma-Aldrich), permeabilized with PBS containing 0.5% Triton X-100 (Sigma-Aldrich) and blocked with PBS contained 1% BSA (Sigma-Aldrich). Then the cells were stained with monoclonal antibody (mAb) against PRRSV-N protein (Clone No. 6D10, made in-house) as previously described [30]. Specific binding between the antibody and its target was detected using Alexa Fluor^®^488-labeled goat anti-mouse IgG (Thermo Fisher Scientific, Waltham, MA, USA). Cellular nuclei were counterstained by 4,6-diamidino-2-phenylindole (DAPI) (Thermo Fisher Scientific) at 37 °C for 10 min and observed under a Leica DM1000 fluorescence microscope (Leica Microsystems, Wetzlar, Germany). All images were captured and processed using Leica Application Suite X (Version 1.0. Leica Microsystems). IFA-positive cells from representative images were quantified using ImageJ software (Version 1.52a, National Institute of Health, Bethesda, MD, USA)

### 2.5. Reverse Transcription and Quantitative PCR (RT-qPCR)

Total RNA was extracted from cells using TRizol Reagent (Thermo Fisher Scientific) and reverse-transcribed using the PrimeScript RT reagent Kit (TaKaRa, Dalian, China) following the manufacturer’s instructions. The qPCR was conducted using 2×RealStar Green Fast Mixture 2× PCR (GenStar, Beijing, China). Transcripts of the β-actin were also amplified and used to normalize total RNA input. The relative quantification of target gene expression was calculated with the 2^−ΔΔCt^ method. Primers used for qPCR and their sequences are listed in Table 1.

### 2.6. Western Blot Analysis

Cells were lysed by 1× Laemmli sample buffer (Bio-Rad Laboratories, Hercules, CA, USA) prior to SDS-PAGE. Proteins were separated using 12% SDS-PAGE gels, followed by transferring to a PVDF membrane as previously described [31]. After blocking, membranes were probed with homemade Mab-6D10 against the PRRSV-N protein and Mab against β-actin (Abcam, Cambridge, MA, USA). Specific binding between antibodies and their targets was detected using HRP-conjugated goat anti-mouse IgG (Thermo Fisher Scientific) and revealed with ECL substrate (Beyotime, Jiangsu, China). Chemiluminescence signal acquisition was conducted using a ChemiDoc MP Imaging System (Bio-Rad Laboratories) and analyzed using ImageLab software (Version 5.1, Bio-Rad Laboratories).

### 2.7. Analysis of Virus Attachment and Internalization

MARC-145 cells or PAMs were pre-incubated with the 40 kDa linear PEI for 1 h at 37 °C or left untreated (control), followed by pre-chilling at 4 °C for 30 min prior to PRRSV inoculation. The PRRSV-JXA1 strain was used to inoculate the cells at 4 °C for 1 h at the indicated MOI. Similarly, pre-chilled MARC-145 cells or PAMs were co-incubated with a mixture of PRRSV-JXA1 virus and the 40 kDa linear PEI at 4 °C for 1 h or PRRSV-JXA1 virus only. Subsequently, the cells with the indicated treatment were washed three times with cold PBS to remove unbound virions before being harvested for RT-qPCR to evaluate viral RNA level from the virions bound.

For the virion internalization assay, PRRSV-JXA1 at the indicated MOI was used to inoculate MARC-145 cells or PAMs at 4 °C for 1 h to allow the sufficient attachment of virions to the cell surface without triggering endocytosis as previously instructed [32]. After removing unbound viral particles by washing cells with cold PBS for three times, fresh medium containing the 40 kDa linear PEI was used to treat MARC-145 or PAMs followed by shifting cells to 37 °C to trigger endocytosis, which allows the internalization of bound PRRSV virions on the cell surface. One hour later, the cells were washed with cold PBS again and further incubated with 50 μg proteinase K (Sigma-Aldrich) for 45 min at 4 °C to remove non-internalized virions located on the cell surface. After the inactivation of proteinase K by a protease inhibitor cocktail (Roche, Basel, Switzerland), the cells were harvested for RT-qPCR analysis of the viral RNA of internalized PRRSV virions.

### 2.8. Statistical Analysis

All experiments were conducted with at least three independent replicates. The results were analyzed using GraphPad Prism version 5.0 (GraphPad Software, San Diego, CA, USA). Statistical significance was determined by student’s *t*-test if two groups were compared. A *P* value less than 0.05 was considered statistically significant.

## 3. Result

### 3.1. Evaluation of Cytotoxicity of Polyethylenimine in MARC-145 and PAM Cells

In this study, three different forms of Polyethylenimine (PEI) were studied for their effect on PRRSV replication in both the continuous cell line MARC-145 and the primary PAM cells. They are two linear PEIs (40 and 25 kDa) and one branched PEI (75 kDa). Their chemical structures are illustrated in Figure 1A–C. The cytotoxicity assay showed that no obvious adverse effect in MARC-145 cells was observed for all three forms of PEI up to 10 µg/mL (Figure 1D). However, in the PAMs, the branched PEI was less toxic than the linear PEI, while the 25 kDa linear PEI was more toxic than the 40 kDa PEI (Figure 1E).

Meanwhile, it is notable that the cytotoxic effect of PEI in these two types of cells was different. In MARC-145 cells treated with branched or the 25 kDa linear PEI, a slight increase in cell viability was observed along with an increasing dose of PEI (Figure 1D). On the contrary, the linear PEI had a cytotoxic effect in PAMs along with the incremental concentration. The cell viability of PAMs was significantly impaired by the 40 kDa linear PEI at a concentration higher than 8 μg/mL and by the 25 kDa PEI at a concentration as low as 1 μg/mL. However, it appears branched PEI was non-toxic to PAMs at the concentrations within the 10 μg/mL range (Figure 1E). The CC_50_ of different PEIs in PAMs were calculated to be 7.389, 8.273 and 23.831 μg/mL for the 25 kDa linear PEI, the 40 kDa linear PEI and the 75 kDa branched PEI, respectively, (Appendix A). Based on the cytotoxicity assay, the 40 kDa linear PEI and the 75 kDa branched PEI were selected for the following experiments in MARC-145 cells and PAMs.

### 3.2. The 40 KDa Linear PEI Effectively Inhibits PRRSV Replication in MARC-145 Cells

We next analyzed the effect of the 40 kDa linear PEI (hereby and thereafter, PEI-linear) and the 75 kDa branched PEI (hereby and thereafter: PEI-branch) on PRRSV replication in MARC-145 cells with a range of concentrations from 1 to 8 μg/mL based on the cytotoxicity assay. IFA results showed that the PEI-linear led to a dose-dependent inhibition of PRRSV-SD16 infection, as evidenced by a reduction in numbers of PRRSV-positive cells, whereas only weak inhibition could be observed in MARC-145 cells treated with PEI-branch at a concentration of 8 µg/mL (Figure 2A). By using ImageJ to analyze the IFA-positive cells, the median effective concentration (EC_50_) of the PEI-linear and PEI-branch was calculated to be 2.435 and 6.883 μg/mL, respectively (Appendix A).

To further evaluate the inhibitory effect of both PEIs on PRRSV replication in MARC-145 cells, the level of the nucleocapsid (N) protein in virus-infected cells was determined by Western blot. Similar to the trends of the IFA results, an incremental inhibition of N expression was correlated with an increasing concentration of PEI, and N was below the detection level in cells treated with linear-PEI at the concentration of 6 μg/mL. In contrast, PEI-branch had no significant inhibitory effect on PRRSV replication in MARC-145 cells until the concentration was increased to 8 μg/mL (Figure 2B). Moreover, if PRRSV virus was used to inoculate the cells at 0.1 MOI but treated with a similar PEI dose, the PEI-linear at the concentration of 4 μg/mL could block the PRRSV replication (Figure 2C). Meanwhile, the significant inhibition of PRRSV replication by PEI-branch could be observed at the concentration of 6 μg/mL (Figure 2C).

To further confirm and quantify the effect of PEI on PRRSV, the progeny virions from the cell culture supernatant of MARC-145 cells infected with PRRSV with treatment of two forms of PEI were titrated (Figure 2D). Consistent with the Immunofluorescence assay (IFA) and Western Blotting (WB) results, linear PEI treatment resulted in a 1.5 Log_10_ reduction of infectious virus particles in the medium. Therefore, compared to PEI-linear, the inhibition of PRRSV by PEI-branch was much weaker. In summary, the results indicate that the 40 kDa linear PEI exerted a stronger inhibition on PRRSV replication than the PEI-branch.

### 3.3. The PEI-linear Demonstrates Broad inhibition of Heterogeneous PRRSV-2 Isolates

As a chemical reagent, the inhibitory effect of PEI-linear on PRRSV is expected to be broad spectrum. We next tested whether this PEI could inhibit other *PRRSV-2* isolates in MARC-145 cells. Our results showed that the replication of other *PRRSV-2* isolates (JXA1, GD-HD, and VR-2332) was significantly inhibited by PEI in a dose-dependent manner (Figure 3A–C). Moreover, the progeny virus from cell culture supernatants were titrated. Based on our data, it appears that PEI-linear demonstrated the strongest inhibition to PRRSV-2 VR-2332 for at least two Log_10_ reductions of infectious viral particles (Figure 3D). Notably, the NADC30-like isolate replicated poorly in MARC-145 with a viral titer below 10^5^ TCID_50_/mL (Figure 3D). The inhibitory effect of PEI against different PRRSV strains was not uniform as the titers of certain viral isolates such as JXA1 were higher than the others. Taken together, these data showed that PEI-linear could effectively inhibit several PRRSV-2 isolates tested in MARC-145 cells.

### 3.4. The PEI-Linear Blocks PRRSV Replication in PAMs

Since PAMs are the primary target cells for PRRSV infection in vivo, we also evaluated the antiviral activity of PEI-linear in PAMs. According to the cell viability assay results (Figure 1), PEI-linear at a concentration of 6 μg/mL was used in PAMs. As our preliminary results demonstrated the attachment and infection efficiency of PRRSV to PAMs is nearly as high as 1000-fold in comparison to MARC-145 cells [33]. Therefore, 10^6^ PAMs were inoculated with the PRRSV-2 strains: SD16, JXA1, GD-HD, and NADC30-like, at 0.01 MOI due to the high susceptibility of PAMs to PRRSV. Our results demonstrated that PEI inhibited the replication of the heterogeneous PRRSV isolates in PAMs (Figure 4A). Meanwhile, the RT-qPCR results confirmed the PEI inhibition of PRRSV in PAMs as well (Figure 4B). Furthermore, the titration results showed that PEI-linear inhibited all the PRRSV isolates tested (Figure 4C). Interestingly, it appears that the NADC30-like isolate replicated similarly in PAMs as the other isolates (Figure 4C). Taken together, these data demonstrated that PEI-linear inhibits PRRSV infection in PAMs as well.

### 3.5. The PEI-Linear Blocks the Attachment of PRRSV Virions but Not Internalization in MARC-145 Cells

It has been reported that PEI mediated the blockage of the attachment of HPV and HCMV to surface of susceptible cells and this probably occurs through competitively binding the same attachment sites on cells between PEI and virions [26]. Thus, we next determined whether PEI prevents the attachment of PRRSV virions to MARC-145 cells. We first tested whether the inhibition of PRRSV infection by PEI was due to the reduced virion attachment. After the inoculation of MARC-145 cells with the virus and PEI mixture, the cells were maintained at 4 °C without triggering endocytosis but allowed virion attachment. Then, the attached virions were quantified by RT-qPCR. The results showed that PEI reduced the PRRSV virion attachment to MARC-145 cells at both 5 and 10 MOI when PEI and the virus were added at the same time (Figure 5A). To further confirm this result, we pre-incubated cells with PEI-linear first, followed by inoculating the cells with PRRSV (5 or 10 MOI) at 4 °C. The RT-qPCR analysis suggests that there was reduced binding of viral particles to MARC-145 cells as well (Figure 5B).

To determine whether the PEI affects PRRSV internalization, we inoculated MARC-145 cells with the virus first for attachment at 4 °C, followed by PEI treatment and temperature shift to 37 °C to trigger endocytosis-mediated internalization of the virions. The viral particles remaining on cell surface were removed by treatment with protease K. So only internalized virions were harvested for qPCR. The data suggest that PEI could not block the internalization of attached virions as shown by similar viral RNA levels in both PEI-treated and mock-treated cells (Figure 5C). Taken together, these data suggest that the 40 kDa linear PEI inhibits PRRSV via blocking virion attachment to MARC-145 cells.

### 3.6. The 40 kDa Linear PEI Blocks the Attachment of PRRSV Virions to PAMs via Acting on the Virus but Not Cells

Since PAMs are the major target of PRRSV in vivo, we investigated whether PEI has a similar inhibition mechanism to PRRSV in PAMs as that of MARC-145 cells. Similar to MARC-145 cells, thw co-administration of PRRSV and PEI to PAMs reduced the binding of PRRSV virions to PAMs (Figure 6A). Moreover, it is notable that the susceptibility of PAMs to PRRSV is much higher than MARC-145 cells, since the inoculation of PRRSV at 0.01 MOI to PAMs was sufficient for RT-qPCR analysis of attached virions but it required at least an MOI of 5 of the virus to inoculate the MARC-145 cells for RT-qPCR analysis of attached virions (below the detection limit in MARC-145 cells if below 1 MOI). Meanwhile, if PAMs were pre-incubated with PEI-linear, followed by inoculation with PRRSV-JXA1 at 4 °C, there was a significant increase in virion binding compared with PAMs without PEI pretreatment (Figure 6B), which is different from that of MARC-145 cells. Moreover, if PAMs were inoculated with PRRSV before PEI treatment, followed by a temperature shift to trigger endocytosis, the analysis of the internalized PRRSV virions showed that PEI also promoted the internalization of attached virions as well (Figure 6C). Therefore, the data suggest that PEI-linear may act on the virus rather than cells to prevent the binding of the virus to PAMs. Meanwhile, the pre-treatment of PAMs with PEI even promotes PRRSV attachment and internalization to PAMs. Taken together, these data suggest that the 40 kDa PEI inhibits PRRSV attachment and is capable of conferring the broad inhibition of heterogeneous *PRRSV-2* isolates.

## 4. Discussion

PEI, also known as polyaziridine, is a polymer with a repeating structure that is composed of the amine group along with a two-carbon aliphatic CH_2_CH_2_ spacer. It was reported that PEI could be in linear, branched, and dendrimeric forms [34], while only linear and branched molecules of PEI are available from commercial suppliers. PEI has been used for many applications, usually due to its polycationic character. For cell biology research, PEI has been used as a transfection reagent [35]. Also, PEI is explored for its antiviral activity. For HPV, the 25 kDa linear PEI blocks the primary attachment of HPV16 and HCMV to susceptible cells [26]. It appears that PEI preincubation with cells blocks HPV and HCMV binding to their primary receptor HSPG [26]. Moreover, the antiviral activity of PEI has been investigated in vivo as well. Intranasal administration of the 25 kDa linear PEI suppresses influenza virus infection in mice [36]. The 3610 Da branched PEI combined with liposomes was shown to strongly enhance antiviral efficiency against herpes simplex virus type 2 in a mouse model [37].

Currently, the inhibition mechanism of PEIs against viral infections is still not fully understood and appears to depend on blocking virion attachment to susceptible cells. As a result, PEI potentially interferes with electrostatic interactions between viral surface proteins and host receptors [26]. However, it was reported that branched PEI with a molecular mass of 70 kDa accelerated HIV-1 infection despite its inhibition of HIV-1 attachment to cells [27], which may be partially due to the promotion of the exposure of co-receptor and/or viral entry into cells via the influence of cell membrane fluidity [27].

In our study, the 40 kDa linear PEI was shown to be effective against PRRSV infection through the inhibition of PRRSV virion attachment to MARC-145 cells, the most frequently used cell line for PRRSV study in vitro. Meanwhile, PEI treatment of MARC-145 cells does not affect the internalization of attached PRRSV virions on the cell surface. This inhibition mechanism appears to be consistent with a previous observation for HPV and HCMV [26]. Conversely, the inhibition of PRRSV in PAMs by the PEI was weaker than that in MARC-145 cells when the same dose of PEI was used. Notably, based on our data, although PEI prevented the attachment of PRRSV virions to PAMs when the PEI and virus were co-administrated simultaneously, further analysis suggests that the pre-treatment of PAMs with PEI enhanced the attachment of PRRSV virions. Moreover, PEI enhanced the internalization of PRRSV virions, which appears to be similar to the scenario in HIV but different from PRRSV in MARC-145 cells. Therefore, these data suggest that the exact procedures of the attachment and internalization of PRRSV virions in MARC-145 cells and PAMs may be different, which could also be supported by the different susceptibility of both cells to PRRSV as well. Moreover, how such a difference affects PRRSV pathogenesis or antibody-mediated viral neutralization (mainly via blocking virion attachment to susceptible cells) remains unclear. Meanwhile, when acting as a transfection agent, PEI condenses DNA into positively charged particles, which bind to anionic cell surface residues to deliver the PEI/DNA complex into the cells via endocytosis [38,39]. Once inside the endosome, amines of PEI result in an influx of counter-ions and a lowering of the osmotic potential. Therefore, osmotic swelling disrupts the vesicle to release the PEI/DNA complex into the cytoplasm [38,39]. It has been known that PRRSV virions enter PAMs via clathrin-mediated endocytosis [32]. It is possible that PEI promotes endocytosis after PRRSV virion attachment or disrupts the endosome vesicle to release PRRSV RNA into the cytoplasm in PAMs to enhance the virion internalization. However, why such a scenario in PAMs is different from MARC-145 cells remains unclear.

On the one hand, PK-15 cells (immortalized swine kidney cell line is not susceptible for PRRSV) stably expressing porcine CD163 (the essential receptor for PRRSV infection) support the complete PRRSV replication cycle [40]. On the other hand, PK-15 cells stably expressing CD169 (a putative receptor involved in PRRSV internalization) support the internalization of PRRSV virions but not the uncoating of nucleocapsid and fusion with the endocytic vesicle membrane [41]. These data imply that the endocytosis-mediated internalization and uncoating procedure of PRRSV particles in PAMs and immortalized cell lines such as PK-15 and MARC-145 is different. Such a difference may contribute to the different inhibitory effect of the 40 kDa linear PEI on PRRSV virions.

The cytotoxicity of different PEI forms was evaluated in this study as well. A previous report suggested that branched (25 kDa) and linear (750 kDa) PEI can both induce membrane damage and initiate apoptosis in human cell lines [42]. Based on our observation, the cytotoxicity of PEI appears to be cell type and structure dependent. Both linear PEIs demonstrated better tolerance in MARC-145 cells than primary PAMs, while branched PEI had a much better tolerance than linear PEI in PAMs. However, the reason for the differences remains elusive. This implies that a certain modification or optimization of PEI molecular structure may be explored to reduce cytotoxicity while maintaining comparable antiviral activity.

Mechanically, PEI blocks the binding of virus to its receptors to prevent the attachment of virions to susceptible cells as demonstrated for HPV16 and HCMV [26]. This is also consistent with our observation that PEI prevents PRRSV attachment to MARC-145 cells. The difference in PEI inhibition of PRRSV in MARC-145 and PAM cells suggests that the interaction between PRRSV virions and receptors in these two types of cells might be different. Several membrane proteins have been identified as potential receptors for PRRSV infection in permissive cells, such as heparin sulfate (HS) [9], vimentin [10], CD151 [11], porcine CD163 [12], sialoadhesin (CD169) [13], DC-SIGN (CD209) [14] and MYH9 [15]. However, how these receptors cooperate to mediate the PRRSV infection remains elusive.

For HIV, branched PEI enhanced HIV-1 infection partially due to its promotion of the exposure of co-receptor and/or viral entry into cells via the influence of cell membrane fluidity [27]. Since PRRSV might use several receptors in PAMs in vivo, it is possible that PEI may promote the exposure of co-receptor or receptors and influence the cell membrane fluidity of PAMs.

Moreover, it is also notable that the inhibitory effect of the 40 kDa linear PEI on different *PRRSV-2* isolates is variable, especially in MARC-145 cells. This is somehow unusual as the PEI inhibition of PRRSV should be non-specific and such an inhibition on different PRRSV isolates should be similar. It is unclear whether PEI blocks the interaction of PRRSV virions with the receptors mentioned above indiscriminately or whether PEI preferably blocks the interaction of PRRSV virions with certain receptors from the list above. It is possible that there is a different receptor preference among various PRRSV isolates, as evidenced by the observation that the replacement of the fifth scavenger receptor cysteine-rich domain (SRCR5) of porcine CD163 with the SRCR5 domain of the human CD163-like homolog (CD163Li) only confers resistance to PRRSV-1 but not to PRRSV-2 [43]. Therefore, it is possible that, due to an uneven PEI blockage to the interaction of PRRSV virions to different receptors along with a strain-specific receptor preference of various PRRSV isolates, the PEI cannot confer a consistent inhibition among different PRRSV isolates. Moreover, it appears that different steps and variable outcomes of PEI mediated inhibition between MARC-145 cells and PAMs suggest that there is a difference in the attachment and internalization of PRRSV virions on these two cells. The impact of such a difference on our understanding of PRRSV virion attachment and internalization as well as virus neutralization and pathogenesis in vivo require further investigation.

In conclusion, the 40 kDa linear PEI demonstrates an inhibitory effect against different PRRSV-2 isolates both in MARC-145 cells and PAMs with tolerable cytotoxicity. Further study is needed to investigate whether PEI can confer the inhibition of PRRSV infection in swine. Meanwhile, the potential different internalization procedure of PRRSV in PAMs and MARC-145 cells warrants further investigation.

## Figures and Tables

**Figure 1 viruses-11-00876-f001:**
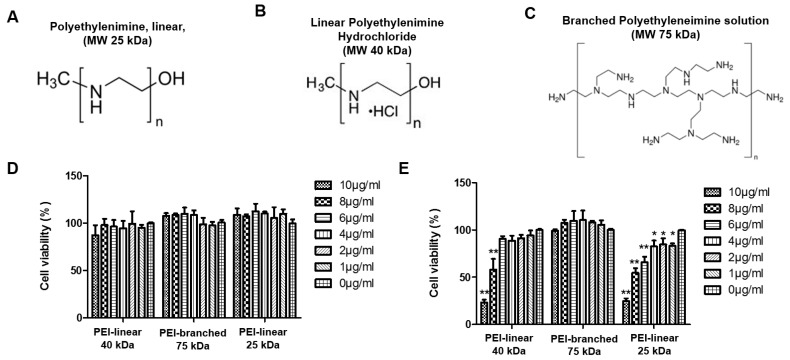
Illustration of the structures of different molecules of polyethylenimine (PEI) and the evaluation of in vitro cytotoxicity. (**A**) Structure of linear PEI with a molecular weight of 25 kDa; (**B**) Structure of linear PEI-hydrochlorides with a molecular weight of 40 kDa; (**C**) Structure of branched PEI with a molecular weight of 75 kDa; (**D**) Cell viability assay of MARC-145 cells incubated in the presence of PEI at the indicated concentrations at 37 °C for 24 h. (**E**) Cell viability assay of pulmonary alveolar macrophage (PAM) cells incubated in the presence of PEI at the indicated concentrations at 37 °C for 24 h. All experiments were repeated at least three times, and the data were presented as the mean ± SD, which is further subjected to Student’s *t*-test. Significant differences between indicated groups were marked by “*” (*P* < 0.05) and “**” (*P* < 0.01).

**Figure 2 viruses-11-00876-f002:**
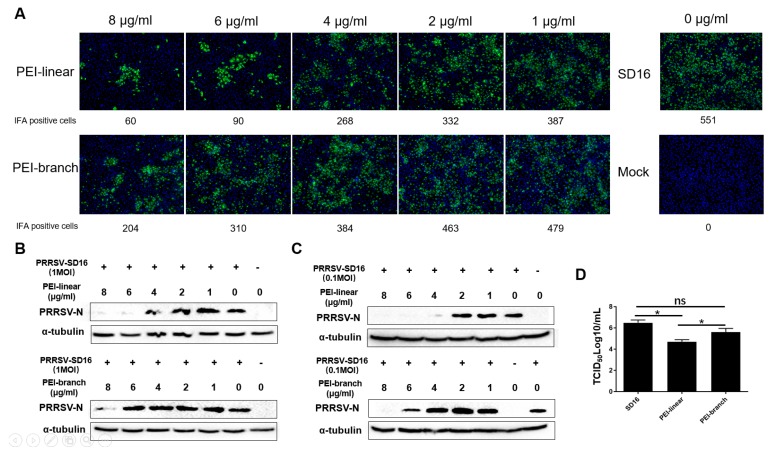
The 40 kDa linear PEI is more effective in inhibiting porcine reproductive and respiratory syndrome virus (PRRSV) replication than the branched PEI. (**A**). Immunofluorescence assay (IFA) of MARC-145 cells that were infected with PRRSV-SD16 in the presence of PEI. The virus at an MOI of 1 was diluted in DMEM and mixed with 40 kDa PEI-linear or PEI-branch at the indicated concentrations for 1 h before being inoculated to the cells. IFA with anti-PRRSV-N Mab-6D10 was done 24 hpi. MARC-145 cells infected with PRRSV-SD16 alone and non-infected cells were included as controls. Quantification of IFA-positive cells was conducted using ImageJ software. (**B**). Western blotting of MARC-145 cells that were infected with PRRSV-SD16 at an MOI of 1 in the presence of PEI. (**C**) Western blot of MARC-145 cells that were infected with PRRSV-SD16 at an MOI of 0.1 in the presence of PEI. (**D**) Progeny PRRSV virions from the cell culture supernatant of MARC-145 cells that were infected with PRRSV-SD16 at an MOI of 0.1 in the presence of PEI at a concentration of 6 μg/mL. Cell culture supernatants were harvested at 24 hpi and titrated for TCID50/mL. All experiments were repeated at least three times, and the data were presented as the mean ± SD along with subjection of Student’s *t*-test. Significant differences between indicated groups were marked by “*” (*P* < 0.05). No significant difference is marked as “ns”.

**Figure 3 viruses-11-00876-f003:**
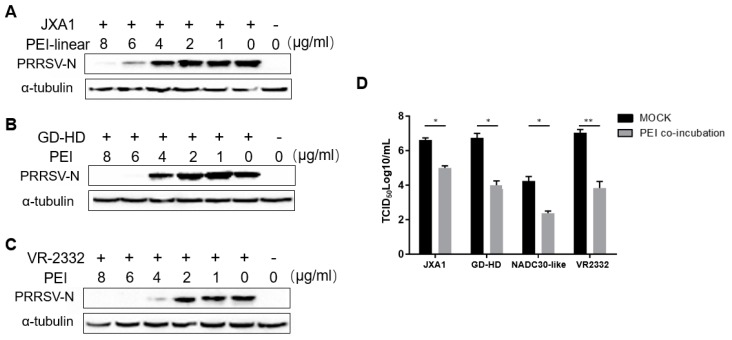
PEI demonstrates broad inhibition of heterogeneous *PRRSV-2* isolates in MARC-145 cells. MARC-145 cells were incubated with a mixture containing 1 MOI of PRRSV JXA1 (**A**), GD-HD (**B**) and VR-2332 (**C**) mixed with the 40 kDa linear PEI at the indicated concentrations for 1 h. Then the cells were washed and cultured for an additional 24 h. Replication of PRRSV was determined in Western blot with Mab-6D10 against N protein. (**D**) PRRSV yields in MARC-145 cells infected with the mixture of different PRRSV virus strains at 0.1 MOI and PEI-linear (6 μg/mL). The mixture was removed and replaced by a fresh medium. The cells were incubated for 24 h before the cell culture supernatant was harvested for titration. All experiments were repeated at least three times, and the data were presented as the mean ± SD, which is further subjected to Student’s t-test. Significant differences between indicated groups were marked by “*” (*P* < 0.05) and “**” (*P* < 0.01).

**Figure 4 viruses-11-00876-f004:**
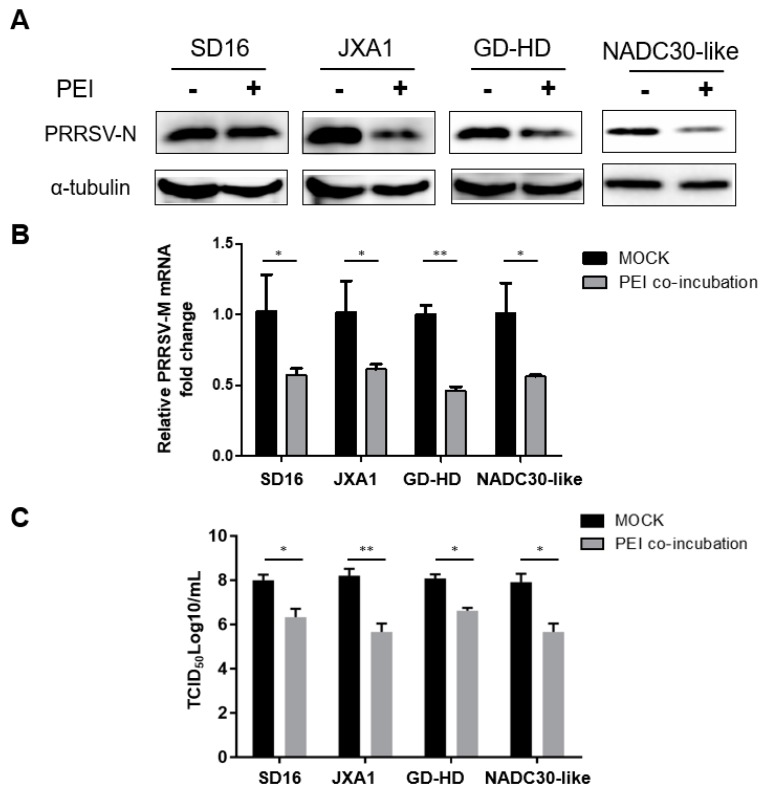
PEI-linear inhibits the replication of PRRSV-2 isolates in PAMs. (**A**). PAMs were inoculated with the virus and PEI mixture containing the indicated PRRSV strains (SD16, JXA1, GD-HD, NADC30-Like, 0.01 MOI) and PEI-linear 6 μg/mL for 1 h. The cells were then washed with PBS buffer and incubated for another 24 h. PRRSV N level was determined with Western blot. (**B**). PRRSV RNA levels in PAMs inoculated with the virus and PEI mixture containing the indicated PRRSV strains (0.01 MOI) and 6 μg/mL linear PEI. (**C**) Progeny PRRSV virions in cell culture supernatant of PAMs infected with different PRRSV strains at an MOI of 0.01 in the presence of linear PEI (6 μg/mL) for 24 h. All experiments were repeated at least three times, and the data were presented as the mean ± SD, which is further subjected to Student’s *t*-test. Significant differences between indicated groups were marked by “*” (*P* < 0.05) and “**” (*P* < 0.01).

**Figure 5 viruses-11-00876-f005:**
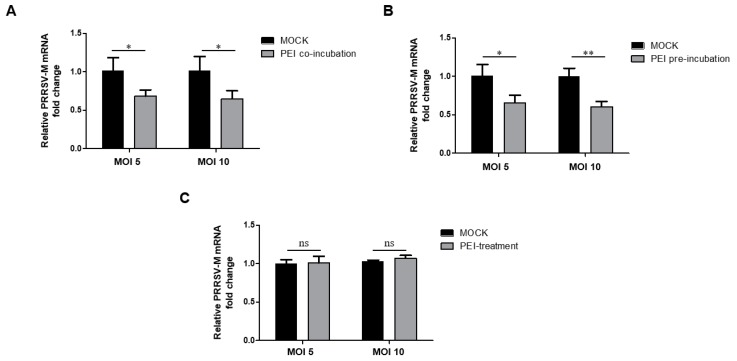
PEI inhibits PRRSV virion attachment to MARC-145 cells but not virion internalization. (**A**). PEI inhibits PRRSV virions attachment to cells when co-incubated with PRRSV to MARC-145 cells. PRRSV RNA levels in MARC-145 cells that were incubated with a pre-chilling virus and PEI mixture containing an MOI of 5 or 10 PRRSV-JXA1 strain mixed with 6 μg/mLPEI-linear for 2 h at 4 °C. Then, the cells were harvested for RT-qPCR analysis. MARC-145 cells inoculated with the virus only without PEI were included as a control. (**B**). Pre-incubation of MARC-145 cells with PEI inhibits virion attachment. PRRSV RNA levels in MARC-145 cells that were pre-incubated with 6 μg PEI for 1 h, followed by chilling at 4 °C for 30 min and inoculation with an MOI of 5 or 10 PRRSV-JXA1 strain at 4 °C for 1 h. (**C**). PEI treatment of MARC-145 cells does not affect virion internalization. MARC-145 cells were inoculated with an MOI of 5 or 10 PRRSV-JXA1 for 1 h at 4 °C, followed by washing with cold PBS, treatment with PEI, temperature shift to 37 °C to trigger virion internalization via endocytosis, and removing non-internalized virions with protease K treatment. Then the cells were harvested to evaluate the PRRSV RNA level to determine internalized virions. All experiments were repeated at least three times, and the data were presented as the mean ± SD, which is further subjected to Student’s *t*-test. Significant differences between indicated groups were marked by “*” (*P* < 0.05) and “**” (*P* < 0.01).

**Figure 6 viruses-11-00876-f006:**
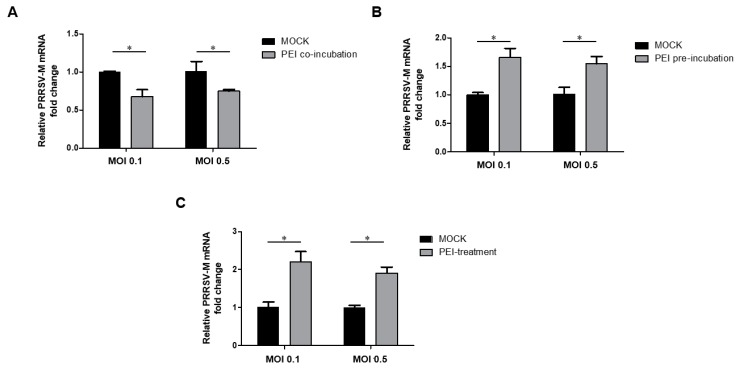
PEI inhibits the attachment of PRRSV virion to PAMs but promotes PRRSV internalization into PAMs. (**A**). PEI prevents virion attachment to cells when co-incubated with PAMs. PRRSV RNA levels in PAMs that were incubated with a pre-chilling virus and PEI mixture containing an MOI of 0.1 or 0.5 PRRSV-JXA1 virus and 6 μg/mL PEI-linear for 2 h at 4 °C to allow attachment. Cells inoculated with the virus only were included as a control. (**B**). Pre-incubation of PAMs with PEI enhances virion attachment to PAMs. PRRSV RNA levels in PAMs that were pre-incubated with 6 μg PEI-linear for 1 h, followed by chilling at 4 °C for 30 mins and inoculation with an MOI of 0.1 or 0.5 of PRRSV-JXA1 virus at 4 °C for 1 h. (**C**). PEI treatment of PAMs enhances virion internalization into PAMs. PRRSV RNA levels in PAMs that were inoculated with an MOI of 0.1 or 0.5 PRRSV-JXA1 for 1 h at 4 °C, followed by washing with cold PBS, treatment with 6 μg PEI and temperature shift to 37 °C to trigger virion internalization via endocytosis. All experiments were repeated at least three times, and the data were presented as the mean ± SD, which is further subjected to Student’s *t*-test. Significant differences between indicated groups were marked by “*” (*P* < 0.05) and “**” (*P* < 0.01).

**Table 1 viruses-11-00876-t001:** Primers and corresponding sequence used for qPCR.

Gene	Forward Primer (5′–3′)	Reverse Primer (5′–3′)
PRRSV-M	TGGGGAGTGTACTCAGCCAT	AATGTACTTGCGGCCTAGCA
β-actin	GGCATCCACGAAACTACCTT	TGATCTCCTTCTGCATCCTG

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
