# Peer review of "The 40 kDa Linear Polyethylenimine Inhibits Porcine Reproductive and Respiratory Syndrome Virus Infection by Blocking Its Attachment to Permissive Cells"

_viruses, 2019, doi:10.3390/v11090876_

Round 1

Reviewer 1 Report

Jie Wang and colleagues demonstrated the inhibitory effect of polyethylenimine (PEI) on the entry of porcine reproductive and respiratory syndrome virus (PRRSV) and suggested that the PRRSV can be internalized differentially in different cell types. In my opinion, the manuscript is well-written and the data are well-presented. Below please find my specific comments:

Major comments:

The authors presented the dose-dependent cytotoxic effect of different PEIs on two cell lines (MARC-145 and PAMs). The discussion of the cytotoxic effect is missing. Did the authors check the pH in the culture medium treated with PEIs? Since the PEI contains high numbers of amine groups, the pH can be changed and thus the cell apoptosis can be induced.

Figure 2A – please quantify the IFA using the available software (i.e. ImageJ). Is it possible that the low concentration of PEI enhance the PRRSV entry?

Figure 2C – there is a specific band for PRRSV-N for the cells which are not infected with PRRSV and not treated with PEI (-PRRSV-SD16, 0 PEI-branch), and no presence of the virus in cells infected with PRSSV and without PEI treatment (+PRRSV-SD16, 0 PEI-branch).

Figure 2D – please change the statistic test to combine three groups. With t-student test, you can compare only control vs treatment, and since the authors compared the activity of linear and branched PEI, the appropriate statistical test should be used.

Minor comments:

Page 2 Line 43 – please correct ‘susceptibale’

Page 6 Line 210 – please correct ‘progney’

Figure 3 – the figure’s titles should be bold

Figure 3 – can the authors implement in the manuscript WB of dose-dependent PEI effect on the NAD30-like PRRSV-N expression in the MARC-145 cells? On the Fig. 3D, the PEI-effect on the infectivity is shown for four isolates. Please be consistent.

Figure 3D, Figure 4B – please be consistent with the names of isolates – PRRSV GD or PRRSV GD-HD?

Figure 4 – why the PRRSV-VR2332 isolate was not used in this experiment?

Figure 5 - the figure’s titles should be bold

Figure 5 – in my opinion, it would be useful for the readers if the graphs had small subtitles to easily understand which treatment was used

Figure 5, Figure 6 – please change ‘5MOI’, etc. to ‘MOI 5’, …

Page 10 Line 330 – what does ‘[the ratio of internalization to attachment?]’ mean?

Author Response

Reviewer No.1

Jie Wang and colleagues demonstrated the inhibitory effect of polyethylenimine (PEI) on the entry of porcine reproductive and respiratory syndrome virus (PRRSV) and suggested that the PRRSV can be internalized differentially in different cell types. In my opinion, the manuscript is well-written and the data are well-presented. Below please find my specific comments:

Response: We thank the reviewer for the comment. We have revised the manuscript accordingly. Please see our revised manuscript.

Major comments:

The authors presented the dose-dependent cytotoxic effect of different PEIs on two cell lines (MARC-145 and PAMs). The discussion of the cytotoxic effect is missing. Did the authors check the pH in the culture medium treated with PEIs? Since the PEI contains high numbers of amine groups, the pH can be changed and thus the cell apoptosis can be induced.

Response: We thank the reviewer for these suggestions. The discussion of PEI cytotoxic effect had been added. Please see line 402 to line 409 in the revised manuscript. According the reviewer’s suggestion about the pH, we have checked the pH of cell culture medium (DMEM and RPMI 1640) before and after adding different dose of PEI (from 1 μg to 10 μg into 1 mL medium) and no significant change of pH in medium was observed (determined by Potentiometric pH Sensor). This result is expected since the phenol red as the standard component within either DMEM or RPMI1640 will change its color if there is a drastic fluctuation of pH after PEI addition, which has never been observed throughout the whole study. We hope these data and explanation could satisfy this reviewer.

Figure 2A – please quantify the IFA using the available software (i.e. ImageJ). Is it possible that the low concentration of PEI enhance the PRRSV entry?

Response: We thank the reviewer for this comment, we have qualified the IFA using ImageJ, and the numbers of IFA-positive cells had been labeled under each image of IFA. Please see our new Figure.2A in the revised manuscript. For the issue of low concentrations, it appears that when 1 μg PEI was used to inhibit different PRRSV isolate, there is a slight increase of WB band for SD16 and GD-HD infected MARC-145 cells, but not JXA1 and VR-2332. As PEI could bind to anionic cell surface residues to deliver PEI/DNA complex into the cells via endocytosis, it is possible that low concentration of PEI has a mild stimulation of endocytosis. However, as demonstrated in our study, the PEI’s inhibition towards different PRRSV strains are variable and may relate with numerous cellular receptor preference and cell tropism by different PRRSV strains (please see the discussion parts from line 423 to line 438). Therefore, it is possible that PRRSV entry enhancement by mild stimulation of endocytosis with low concentration of PEI may be PRRSV strain specific.

Figure 2C – there is a specific band for PRRSV-N for the cells which are not infected with PRRSV and not treated with PEI (-PRRSV-SD16, 0 PEI-branch), and no presence of the virus in cells infected with PRSSV and without PEI treatment (+PRRSV-SD16, 0 PEI-branch).

Response: We thank the reviewer for the comment. We are sorry for the wrong label here. It has been corrected. Please see new Figure 2C in the revised manuscript.

Figure 2D – please change the statistic test to combine three groups. With t-student test, you can compare only control vs treatment, and since the authors compared the activity of linear and branched PEI, the appropriate statistical test should be used.

Response: We thank the reviewer for this comment. We checked our results and confirmed data presented in Figure 2D only compared data from two groups for student t-test. To ensure there is no misleading for readers, we marked the compared groups with solid lines. Please see our new Figure 2D.

Minor comments:

Page 2 Line 43 – please correct ‘susceptibale’

Response: We thank the reviewer for this comment. It has been changed. Please see line 43 in revised manuscript.

Page 6 Line 210 – please correct ‘progney’

Response: We thank the reviewer for this comment. It has been corrected. Please see line 217 in revised manuscript.

Figure 3 – the figure’s titles should be bold

Response: We thank the reviewer for this comment. It has been corrected. Please see new Figure.3 in the revised manuscript.

Figure 3 – can the authors implement in the manuscript WB of dose-dependent PEI effect on the NAD30-like PRRSV-N expression in the MARC-145 cells? On the Fig. 3D, the PEI-effect on the infectivity is shown for four isolates. Please be consistent.

Response: We thank the reviewer for the comment. As mentioned in the Material and Methods part (line 84 to 85), the NADC30-like PRRSV isolate was only propagated and titrated in PAMs (not in MARC-145). If this isolate was used to propagate in MARC-145 cells, the viral titer is extremely low (TCID50 is generally below 104) and replication of NADC30-like PRRSV isolate cannot be detected using WB in MARC-145 (Data not shown). However, it replicates very well in PAMs. Therefore, this is why we did not conduct dose-dependent PEI effect on the NADC30-like isolate in MARC-145 cells. Conversely, the case for VR-2332 is completely different. As a classical PRRSV isolate, VR-2332 replication in PAMs is much lower than that in MARC-145 cells, and the replication of VR-2332 is very hard to be detected using WB in PAMs. As a result, the four isolates tested in MARC-145 and PAMs to evaluate the PEI are a little different. We hope this reviewer could understand the situation.

Figure 3D, Figure 4B – please be consistent with the names of isolates – PRRSV GD or PRRSV GD-HD?

Response: We thank the reviewer for this comment. The label has been changed. Please see new Figure 3D and Figure 4B

Figure 4 – why the PRRSV-VR2332 isolate was not used in this experiment?

Response: We thank the reviewer for this comment. Please see our response for the NADC30-like isolate above. Not using PRRSV-VR2332 in PAMs is just like the scenario of not using NADC30-like PRRSV isolate in MARC-145 cells.

Figure 5 - the figure’s titles should be bold

Response: We thank the reviewer for this comment. It has been corrected. Please see the new Figure 5 in the revised manuscript.

Figure 5 – in my opinion, it would be useful for the readers if the graphs had small subtitles to easily understand which treatment was used.

Response: We thank the reviewer for this comment. We have added the subtitle for each graph of Figure 5 in figure legends including Figure 6 to help readers to understand the figure. Please see new Figure 5 and Figure 6 in the revised manuscript.

Figure 5, Figure 6 – please change ‘5MOI’, etc. to ‘MOI 5’, …

Response: We thank the reviewer for this comment. It has been changed. Please see the new Figure 5 and Figure 6 in the revised manuscript.

Page 10 Line 330 – what does ‘[the ratio of internalization to attachment?]’ mean?

Response: We are sorry for the typo here. It has been deleted. Please see line 338 in the revised manuscript.

Reviewer 2 Report

The manuscript by Jie Wang et al. describes antiviral and cytotoxicity properties of three structurally divergent polyethylenimine-based inhibitors (PEI) of porcine reproductive and respiratory syndrome virus (PRRSV) in both two permissive cell lines using multiple in vitro assays. The results show, that the compounds are well tolerated in MARC-145 cells; however only one of the three compounds tested showed a cytotoxicity > 10 ug/mL in primary procine pulmonary alveolar macrophages. The antiviral activity was observed to be relatively week and does not result in a complete inhibition of the virus replication in infected cell culture. Despite an interesting topic and well-designed methodology, the work completely lacks a quantitative evaluation of antiviral and cytotoxic effects, e.g. in terms of the CC50 and EC50 values. The scientific relevance of this work could be enhanced by a demonstration of anti-PRRSV effect in vivo, but due to the low in vitro antiviral activity, no significant in vivo effect can be expected.

Author Response

Reviewer No.2

The manuscript by Jie Wang et al. describes antiviral and cytotoxicity properties of three structurally divergent polyethylenimine-based inhibitors (PEI) of porcine reproductive and respiratory syndrome virus (PRRSV) in both two permissive cell lines using multiple in vitro assays. The results show, that the compounds are well tolerated in MARC-145 cells; however only one of the three compounds tested showed a cytotoxicity > 10 ug/mL in primary procine pulmonary alveolar macrophages. The antiviral activity was observed to be relatively week and does not result in a complete inhibition of the virus replication in infected cell culture. Despite an interesting topic and well-designed methodology, the work completely lacks a quantitative evaluation of antiviral and cytotoxic effects, e.g. in terms of the CC50 and EC50 values. The scientific relevance of this work could be enhanced by a demonstration of anti-PRRSV effect in vivo, but due to the low in vitro antiviral activity, no significant in vivo effect can be expected.

Response: We thank the reviewer for these critic comments. To satisfy this reviewer, we calculated the CC50 of different PEIs in PAMs using the available data gained from cytotoxicity assay (Figure S1). For branch PEI, since no obvious cytotoxicity was observed when 10 μg/mL branch PEI was used to inoculate PAMs, the dose of branch PEI was increased up to 40 μg/mL in PAMs for cytotoxicity assay and the data was presented as Figure S2A. Moreover, the CC50 for the branch PEI was calculated as well (Figure S2B). Meanwhile, the EC50 of 40 kDa linear PEI and branch PEI was calculated using the qualified IFA data with ImageJ for Figure 2A, and the data was presented as Figure S3. Based on these new data, the cytotoxicity and inhibition effect of PRRSV replication of PEI appears to be cell type and structure-dependent. This implies that certain modification or optimization of PEI molecular structure may be explored to reduce cytotoxicity of PEI and increase its antiviral activity. We have also discussed the issues in the manuscript. Please see line 403 to 410 of the revised manuscript.

Reviewer 3 Report

In this manuscript, authors described the inhibition of PPRSV by PEI, first, 40 kDa linear PEI and branched PEI were selected for inhibition experiments based on cytotoxic effect. Then the  40 kDa linear PEI was selected for mechanism study in primary and passage cells based on inhibition potency. From the whole, the designs are logical, and results are convincing. Several concerns are listed:

Dose the HCl unit in 40 kDa linear PEImolecule affect the inhibition efficacy? Line191, “40 Kda”should be “40 KDa”. Figure 2C bottom panel, is there something wrong with the WB brand? For the figure 2D, 4C, the Y axile title should be “TCID50Log10/mL” The mixture of PEI and virus was added to cells immediately, No co-incubation?

Author Response

In this manuscript, authors described the inhibition of PPRSV by PEI, first, 40 kDa linear PEI and branched PEI were selected for inhibition experiments based on cytotoxic effect. Then the 40 kDa linear PEI was selected for mechanism study in primary and passage cells based on inhibition potency. From the whole, the designs are logical, and results are convincing. Several concerns are listed:

Response: We thank the reviewer for the comment.

Dose the HCl unit in 40 kDa linear PEI molecule affect the inhibition efficacy?

Response: We believe that the HCL unit plays no role in the inhibition effect for PRRSV. As HCL unit is related with pH of medium, we checked pH of cell culture medium, and there is no significant change of pH in medium used in this study (DMEM or RPMI 1640 with 10% FBS) before and after adding PEI with different doses. It appears HCL unit in 40 kDa linear PEI related to its high solubility in molecular grade water. The solution of 25 kDa linear PEI has to be boiled (as mentioned at line 91 to 92, section 2.2) so the PEI powder could be completely dissolved. Moreover, 25 kDa linear PEI solution has to be maintained at room temperature (as mentioned at line 93 to 94, section 2.2) and precipitation will be observed if stored at 4℃ or -20℃. 

Line191, “40 Kda”should be “40 KDa”.

Response: We thank the reviewer for this suggestion. It has been changed. Please see line 193 in the revised manuscript.

Figure 2C bottom panel, is there something wrong with the WB brand?

Response: We are sorry for the wrong label made here, we have changed the label. Please see the new Figure 2C in the revised manuscript.

For the figure 2D, 4C, the Y axile title should be “TCID50Log10/mL”

Response: We thank the reviewer for this suggestion. We have changed the label.

The mixture of PEI and virus was added to cells immediately, No co-incubation?

Response: Yes, the mixture was added immediately without co-incubation time of PEI and virus for PRRSV inhibition assay, because we wanted to see the immediate effect of PEI on PRRSV.

Round 2

Reviewer 2 Report

I reviewed the revised version of the manuscript: Title: The 40 kDa Linear Polyethylenimine Inhibits Porcine Reproductive and Respiratory Syndrome Virus Infection by Blocking Its Attachment to Permissive Cells Authors: Jie Wang, Jie Li, Nana Wang, Qi Ji, Mingshuo Li, Yuchen Nan, En-Min Zhou, Yanjin Zhang, Chunyan Wu * The manuscript has been significantly improved and now warrants publication in Viruses.

Reviewer 3 Report

no more concerns